# Low-latency time-of-flight non-line-of-sight imaging at 5 frames per second

Ji Hyun Nam [1], Eric Brandt[2,5], Sebastian Bauer [3,5], Xiaochun Liu [1], Marco Renna[4], Alberto Tosi [4], Eftychios Sifakis[2] & Andreas Velten [1,3 ✉]

Non-Line-Of-Sight (NLOS) imaging aims at recovering the 3D geometry of objects that are hidden from the direct line of sight. One major challenge with this technique is the weak available multibounce signal limiting scene size, capture speed, and reconstruction quality. To overcome this obstacle, we introduce a multipixel time-of-flight non-line-of-sight imaging method combining specifically designed Single Photon Avalanche Diode (SPAD) array detectors with a fast reconstruction algorithm that captures and reconstructs live low-latency videos of non-line-of-sight scenes with natural non-retroreflective objects. We develop a model of the signal-to-noise-ratio of non-line-of-sight imaging and use it to devise a method that reconstructs the scene such that signal-to-noise-ratio, motion blur, angular resolution, and depth resolution are all independent of scene depth suggesting that reconstruction of very large scenes may be possible.

[1] Department of Electrical and Computer Engineering, University of Wisconsin – Madison, Madison, WI, USA. [2] Department of Computer Science, University of Wisconsin – Madison, Madison, WI, USA. [3] Department of Biostatistics and Medical Informatics, University of Wisconsin – Madison, Madison, WI, USA. [4] Dipartimento di Elettronica Informazione e Bioingegneria, Politecnico di Milano, Milan, Italy. [5]These authors contributed equally: Eric Brandt, Sebastian Bauer. ✉email: velten@wisc.edu

The ability to image around corners using diffuse relay surfaces has attracted significant interest in the research community after its first analysis[1] and demonstration[2]. It offers applications in many spaces such as autonomous vehicle navigation, collision avoidance, disaster response, infrastructure inspection, military and law enforcement operations, mining, and construction. While other methods have been investigated as well[3–6], robust 3D reconstructions of near room-sized scenes have only been demonstrated using methods that rely on photon Time of Flight (ToF). In ToF methods a short laser pulse is focused on a point $x_p$ on the diffuse relay surface from where the light scatters in many directions. Some photons travel to objects hidden from the direct line of sight and get reflected back to the relay surface. A fast photodetector images a patch $x_c$ on the relay surface recording the time of arrival of photons reflected off $x_c$. Scanning the relay surface point $x_p$ with the laser provides sufficient information for 3D reconstruction of the hidden scene[7]. Recently, ToF NLOS imaging at 1.43 km standoff distance has been presented[8] which underlines the practical applicability of this technique in the future.

Scene reconstruction is possible using a variety of methods[9–13]; see[14] for a thorough review. While fast algorithms have been demonstrated[15–17], these methods require the wall to be scanned in a confocal measurement with a single-pixel sensor that is focused on the same point as the scanned laser resulting in a very low light efficiency due to the small fraction of the relay wall observed by the pixel. To a lesser degree, this method also suffers from the signal contamination from the directly reflected first bounce light. It also requires a fast scan of the surface providing challenges to hardware design. The low light levels and comparably noisy signals of confocal data have so far resulted in capture times of minutes[15,16] or tens of seconds[18] for general scenes which is not acceptable in most applications. The only live ToF NLOS reconstructions to date[15,16] reconstruct retroreflective surfaces that for the specific scenes and using this specialized confocal scanning capture technique provide signals at least 10,000 times higher than diffuse surfaces in the demonstrated geometries and therefore are not indicative of NLOS performance in many real scenes. Supplementary Figure 1 in this work provides a visual comparison between the diffuse and retroreflective objects. The much higher signal levels of retroreflective surfaces have also been discussed in Fig. 1 of the Supplementary Materials to[16].

Several other works have tackled inference and capture speed, but none of these report visible light imaging results of diffuse objects via diffuse relay surfaces with low latency both in terms of light acquisition and reconstruction. Gariepy et al.[19] track a target 30 cm high, 10 cm wide, and 4 cm thick at a distance of roughly one meter from the relay wall. While live reconstruction is possible with the deep neural network presented by Chen et al.[20], the method is only demonstrated on synthetic datasets and experimental data with a confocal scan configuration that requires long acquisition times. A passive approach using a conventional camera exploits shadows cast by edges to provide a real-time angular image around a corner[4], but does not provide 3D reconstructions. Similarly, the approach[6] also uses a conventional camera and exploits indirect reflections to see around a corner; however, the scene consists of an active monitor, and is not passive. Furthermore, no 3D images are provided. Maeda et. al.[21] study NLOS imaging of thermal objects in the infrared spectral range. At these longer wavelengths, many man-made surfaces are flat enough to appear specular, acting like mirrors. Computational reconstruction is not required enabling low latency videos of thermal emitters. A suitable thermal emitter is a human body, but any scene with variations in surface temperature should provide contrast in this wavelength range. Compared to the active, visible light, time of flight imaging scenarios we consider here, the methodology, capabilities, and application range are all quite different. We state our problem as reconstruction via a relay surface that is diffuse (i.e., nearly Lambertian) for the wavelength used. Other approaches that investigate NLOS imaging at different wavelengths include Scheiner et al.[22] who use a Doppler radar to detect and track objects in real-time, but without providing full 3D reconstructions.

Single-Photon Avalanche Diodes (SPADs) have been used successfully as light detector for NLOS imaging, as they are capable of acquiring individual photons with the necessary time resolution[17,18,23]. A central aspect of ToF NLOS is the fact that the light reflected off diffuse objects in the hidden scene returns to the whole relay wall. This aspect and its consequences are explained in detail in the Methods section. The main consequence is that the acquisition time is inversely proportional to the number of SPAD pixels used, meaning that, for example, 10 pixels allow for 10 times faster capture time at the same reconstruction quality. Alternatively, when using many SPAD pixels, the laser power can be reduced by the number of pixels, because the light is being harnessed more efficiently. It is therefore highly beneficial for NLOS imaging to use SPAD array detectors with many pixels along with reconstruction algorithms capable of utilizing their data. In a nutshell, the following analogy between the NLOS imaging and conventional cameras can be drawn: it is very inefficient to acquire an image by point-scanning the scene with a single pixel. Almost all cameras in use today are array detectors, which capture light for all pixels simultaneously, and don't disregard the light that reflects off the other pixels' positions in the scene, as a point detector would do. The same holds for the NLOS imaging setup, all light returning to the relay wall should be harnessed, which then allows for drastically reduced laser power, or alternatively, less noisy reconstructions. Beamforming approaches could potentially be of interest for NLOS imaging, but most likely eventually will also make use of array detectors to capture all the light that returns to the relay wall[24]. Increasing the number of sensor pixels to simultaneously collect light from a larger fraction of the relay surface results in an increase in the captured signal that is proportional to the number of pixels used. However, obtaining a NLOS reconstruction from this non-confocal type of data is more challenging than a reconstruction from confocal data and past algorithms have required run times in the tens of seconds to minutes range for a single frame. While there are commercial SPAD array sensors, existing arrays are poorly suited for NLOS imaging due to their low time resolution, small pixel size, the way data are read out one frame at a time, and the lack of fast gating capability.

It has been speculated that reconstructed scene sizes are limited to a few meters in diameter and objects further from the relay surface would be much harder to reconstruct based on the strong distance dependence of the returned signal strength[14,15].

In this work, we use specifically designed fast-gated NLOS SPAD array detectors alongside our novel reconstruction algorithm designed using the phasor field framework to overcome the deficits in Signal-to-Noise-Ratio (SNR) of NLOS imaging and enable live, low latency NLOS video with depth independent SNR and motion blur. This results in a constant observable motion speed, angular, and depth resolution, and a constant SNR throughout the reconstructed scene. Our proposed real-time NLOS video processing pipeline is illustrated in Fig. 1. The system then can be scaled in the future to use more pixels for further light efficiency that may be used to reconstruct larger scenes at higher resolutions or with larger stand-off distances.

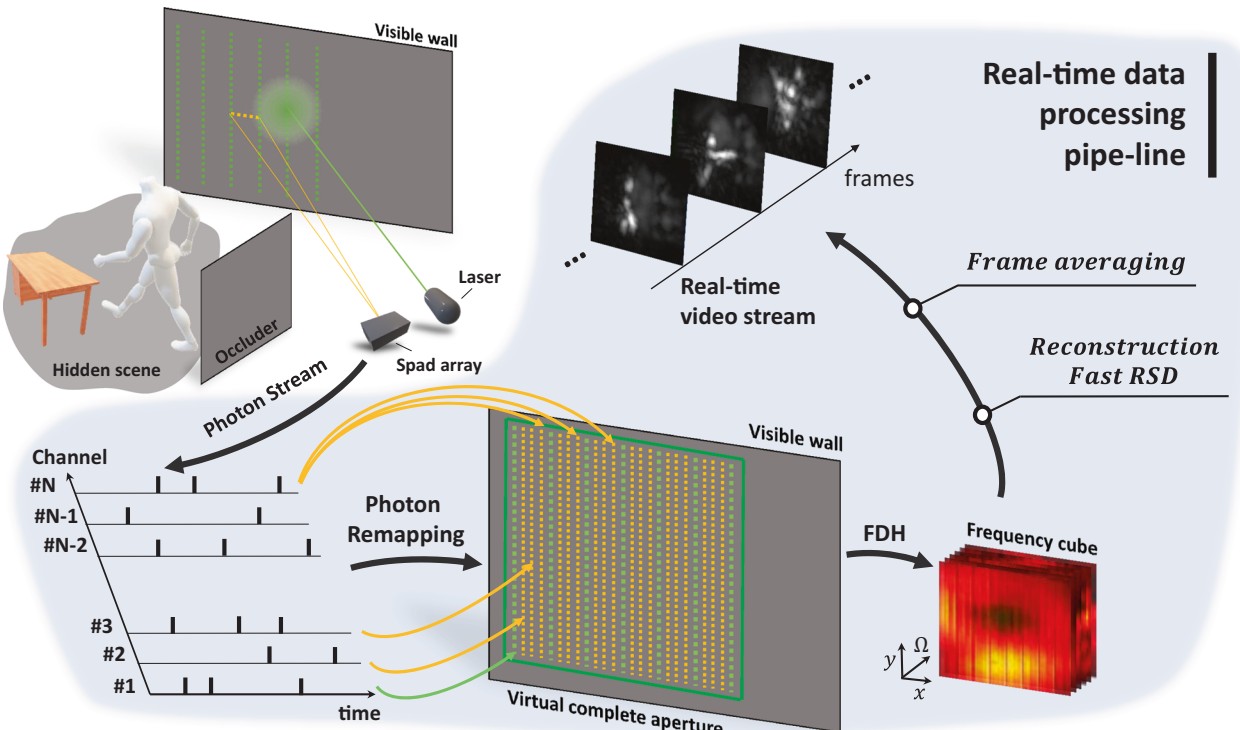

**Fig. 1 Real-time NLOS virtual image processing pipeline.** The imaging system sends the virtual Phasor Field (PF) signal to the visible wall and captures the signal returning from the hidden scene back to the wall. The massive raw photon stream is recorded by the SPAD (Single-Photon Avalanche Diode) array. Raw photons from all channels are virtually remapped into a complete aperture. Then the remapped data is transformed into the frequency domain (Fourier Domain Histogram, FDH) and propagated by the fast Rayleigh Sommerfeld Diffraction (RSD) algorithm. Last, temporal frame averaging yields constant SNR throughout the entire reconstructed volume, and the result is displayed.

## Results

**SNR considerations**. Real-time NLOS videos necessarily constrain the light acquisition period for each frame, which means that the SNR of the data and subsequently the reconstructions have a certain value. Allowing for a longer acquisition period for stationary scenes inevitably increases the SNR, see the Methods section for a theoretical discussion and also Supplementary Fig. 8. The challenge of real-time NLOS imaging, therefore, is to capture as much light as possible during a given short time frame, and SPAD arrays with many pixels solve this problem.

Let us examine the behavior of the NLOS signal as a function of object distance in the scene. Past work states that the signal from a small, fixed-size patch in the hidden scene collected from an individual pair of co-located laser and detector positions $\mathbf{x}_p$, $\mathbf{x}_c$ near the center of the relay surface falls off as $1/r^4$ for the shortest distance $r$ between relay surface and object. However, for the cumulative signal from a complete NLOS measurement comprising a set of $\mathbf{x}_p$ and $\mathbf{x}_c$ this is only true for very large $r$ and does not hold at close distances that apply for most reconstructions. At such close distances, the falloff is smaller than $1/r^4$. An intuitive explanation is this: imagine an infinitely large planar light source (in our case the relay wall illuminated by the laser) that emits a fixed light intensity into one direction. Everywhere in the corresponding half space, we can measure the same intensity, which does not decrease with distance. Otherwise, the total energy would change. A point object very close to a finite relay wall approximately fits this scenario, so the intensity loss from the relay wall to the object is almost negligible. Only the falloff from the object back to the wall has to be taken into account, which follows $1/r^2$. A detailed mathematical analysis of the falloff is provided in Supplementary Note 2. Furthermore, the reduction in resolution at large distances makes considering a patch of fixed

size misleading as the patch simply drops below the resolution limit. Finally, existing investigations consider only the drop in collected signal and ignore the change of noise as a function of $r$. In most conventional optical imaging systems, noise is considered for a fixed angular resolution where the scene area corresponding to an image pixel increases with distance along with the imaging system resolution. For example, in a conventional camera, noise is added to the image by the camera sensor, after image formation has been performed using a lens. By contrast, in an NLOS imaging system, Poisson and sensor noise occur in the measurement before the application of the image formation (i.e., reconstruction) operator. This noise is then propagated through the reconstruction operator which essentially mimics the operation of the imaging lens. As a consequence, the noise in a NLOS reconstruction is different from the noise in a line-of-sight image. In particular, it depends on distance $r$. A detailed derivation of the SNR is also provided in Supplementary Note 3.

**Signal acquisition**. To improve the light efficiency of our setup we use two 16 by 1 pixel fast gated SPAD arrays that were designed specifically for NLOS imaging[25] to image light from a line of patches on the relay surface. A custom-designed SPAD array is necessary because there are no commercial silicon SPAD arrays available yet which simultaneously fulfill all the requirements for good NLOS image reconstructions. Specifically, this array combines a high time resolution of about 50 ps Full Width at Half Maximum (FWHM) with the capability to independently read out each pixel individually to not miss any photon detection and the capability to gate the sensor, i.e., to have it inactive during the period in which the first bounce off the wall occurs. This is crucial for NLOS imaging to prevent this brightest return from overshadowing the subsequent dimmer signals from the hidden scene.

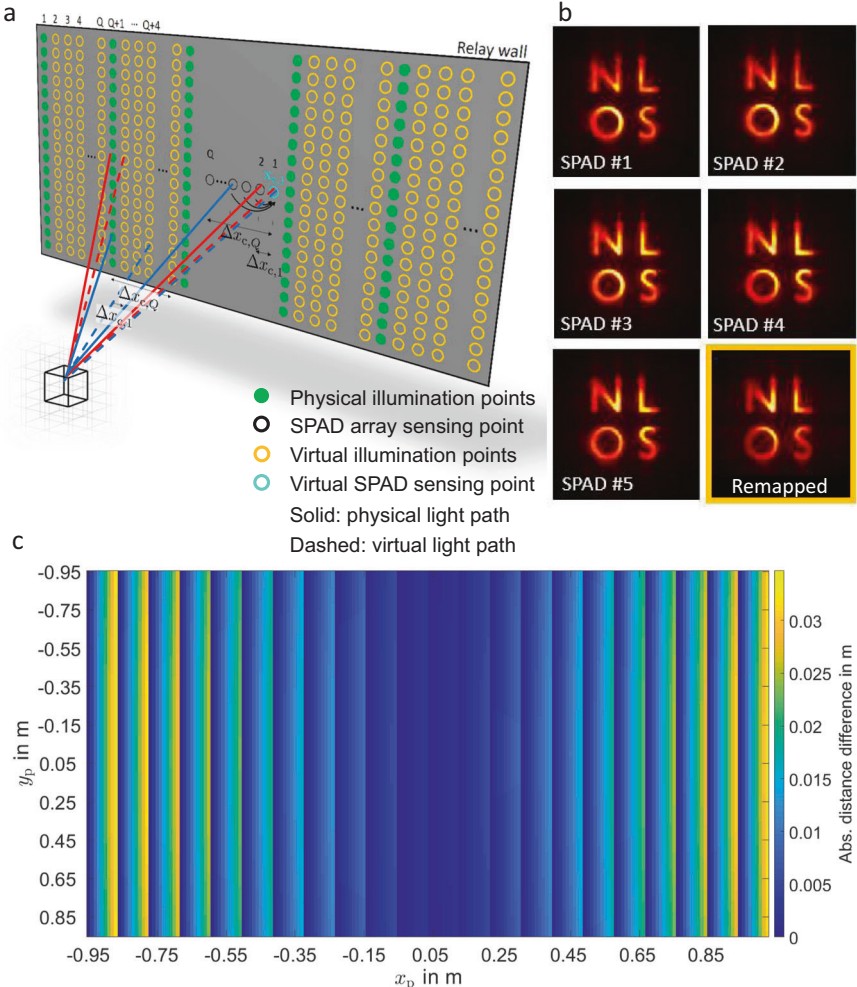

**Fig. 2 Virtual aperture remapping operation. a** Green points are the physically scanned illumination positions and black points are the SPAD array's sensing positions. Data acquired by the SPAD array are remapped into a full virtual aperture; note the highlighted physical and virtual light paths. **b** Five reconstructions of letters from five separate SPAD pixels and one reconstruction using virtually remapped data from five SPAD pixels. c) Absolute difference between physical and virtual light paths depending on the position on the virtual, full grid for the hidden scene voxel location $\mathbf{x}_v = (2\,\text{m},0,0)$. The relay wall parameters are the ones used in this paper; as the absolute path difference for all points is on the order of the system's temporal resolution, the approximation is valid.

In principle, one could scan the relay surface in a continuous grid and use a Phasor Field[17,18,26,27] or filtered backprojection[2] method to perform the reconstruction on the data acquired by SPAD arrays. While there are three different methods to implement a Phasor Field reconstruction, namely simple numerical integration in the primal domain, a backprojection-based solver also in the primal domain, and a diffraction imaging Fourier domain solver based the Rayleigh-Sommerfeld Diffraction (RSD) operator[18,Figure S.1], the first two options, as well as conventional filtered backprojection are not able to reconstruct the scene fast enough for creating real-time videos. The diffraction-based method, can be implemented efficiently (see[17] for a detailed explanation), but requires single-pixel nonconfocal sensors and a dense laser scan[17]. A fix to this is presented in the Methods section where we show that it is possible to remap data captured with a sparse laser scanning pattern and a small SPAD array such that it is equivalent to data captured with a dense laser scan pattern and a single SPAD. This remapping operation makes it possible to simultaneously reduce the demands on the scanner resulting in a higher frame rate, and increase the number of captured pixels increasing the signal strength. The remapping scheme is illustrated in Fig. 2. Throughout this paper, we will only

use Phasor Field reconstruction implemented as fast RSD for the real-time reconstructions, and primal domain Phasor field reconstruction and conventional backprojection for the SNR comparisons.

**NLOS 4D blur kernel**. Let us now examine the application of our capture method and noise model to the reconstruction of large dynamic scenes. It has been shown that motion of the relay surface and thus the virtual camera and physical imaging system can be compensated[28]. We therefore assume that only the motion of objects within the hidden scene can prevent accurate reconstruction. Furthermore, we will assume a maximum object velocity we seek to be able to image.

The spatial resolution of an NLOS image can be described by a three-dimensional Point Spread Function (PSF) $\text{PSF}(x,y,z)$. The widths $\Delta x, \Delta y, \Delta z$ of the PSF along each dimension indicate the resolution of the reconstruction in the different dimensions. It has been shown[16] that the achievable NLOS imaging resolution decreases proportionally with distance. The Phasor Field reconstruction PSF, calculated here by Rayleigh Sommerfeld Diffraction (RSD)[17], matches this increase approximately

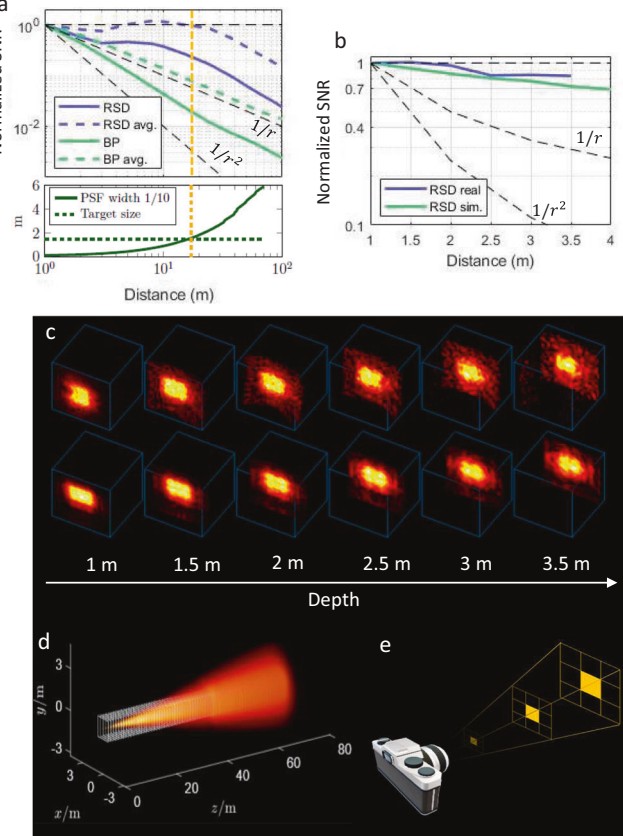

**Fig. 3 NLOS SNR. a** Simulated SNR of Phasor Field reconstruction implemented as fast Rayleigh Sommerfeld Diffraction (RSD) and conventional backprojection (BP) reconstruction of a square target of 1.5 m width, no ambient light, without and with depth-dependent averaging (avg.). **b** Real data as well as simulated results with the ambient light level seen by the current system with room lights on. **c** Reconstructions from real (top) and simulated data (bottom) at different depths. All SNR curves have been normalized to a value of 1 at 1 m to only show the falloff with distance; the absolute values are parameter dependent. **d** Central depth slice of RSD PSF at different depths. The target size is shown as white squares. **e** Pixel size in conventional photography.

following the Rayleigh criterion resulting in a constant angular resolution. This is shown using simulated PSFs in Fig. 3d. As a consequence, the Phasor Field reconstructions are automatically filtered with a depth-dependent low-pass filter that matches the achievable image resolution. This improves the SNR at larger distances compared to other methods.

It is beneficial to express the PSF in spherical coordinates: $PSF(\phi, \theta, r)$ with widths $\Delta\phi, \Delta\theta, \Delta r$ which are largely independent of location in the scene. To incorporate moving objects, we assume that motion blur will only be noticeable when the object in question moves by about the size of the PSF in a given frame. If the motion is smaller the motion blur kernel will be negligible to the PSF and will not significantly affect the reconstruction. This means that for a given maximum object velocity $v$, the required exposure time is proportional to $\Delta\phi/v_\phi, \Delta\theta/v_\theta, \Delta r/v_r$ and increases linearly with distance $r$. Consequently, we can safely average scene voxels at distances far from the relay wall over longer times without expecting visible motion blur. Objects far away from the camera appear to be moving slower. This is not unique to NLOS reconstructions but is the feature of any video. In NLOS reconstructions; however, the depth is known and we therefore have the option to choose a depth-dependent frame averaging in

the reconstruction to take advantage of this effect. We create a reconstruction in which the frame rate is depth dependent. The details of our depth-dependent frame averaging method are described in the Methods section.

**SNR of large scenes.** We use simulated and real captured data to evaluate the SNR of a Phasor Field reconstruction that includes: a) the depth-dependent intensity; b) depth-dependent noise that is the result of passing Poisson and ambient light noise through the reconstruction operator (see Supplementary Note 3 for the details of the noise model); c) the depth-dependent frame averaging along with the depth-dependent band filtering inherent to the Phasor Field algorithm (see Fig. 3). In all cases, we placed a planar diffuse white patch (1.5 m × 1.5 m) in the hidden scene at different depths and collected 100 repeated measurements in the real experiment (depths from 1 m to 3.5 m) and 20000 measurements in the simulated case (depths up to 500 m). Each dataset was processed and reconstructed individually; Fig. 3c shows exemplary reconstructions. All individual reconstruction results are used to evaluate the SNR defined as the mean over the standard deviation at different depths by calculating the sample mean and sample standard deviation over multiple noisy reconstructions. Figure 3a, b show the SNR of the reconstruction at different depths for simulated and real data. For more detailed plots please see Supplementary Note 3.

As we can see in Fig. 3a, the SNR of backprojection (BP) decreases rapidly for large distances, whereas the SNR of the Phasor Field RSD reconstruction decreases slower compared to BP because of the inherent spatial averaging which compensates the SNR loss. Lastly, after applying the optimal depth-dependent frame averaging introduced above we find that the SNR of RSD stays constant approximately up to the distance where the target becomes smaller than the resolution limit of our imaging system. The fact that the resolution decreases with distance is not unique to NLOS imaging, but also occurs in conventional imaging, see Fig. 3d,e.

**Live video results.** Finally, we are ready to present NLOS videos that are both acquired and reconstructed live using the proposed method. To demonstrate real-world capabilities we implemented the optimized pipeline of our proposed method (Fig. 1) and designed a hardware system with two SPAD arrays. See the Methods section for implementation details and hardware specifications. Figure 4 shows several frames of a dynamically moving complex NLOS scene captured and reconstructed live using our system. Our imaging system scans the relay wall with a sparse scan pattern with 190x22 sampling points. The scanning rate is 5 frames per second (fps), hence, the exposure time per frame is 0.2 s. The remapped complete virtual aperture has size 1.9 m × 1.9 m with 190×190 virtual sampling points. The Phasor Field virtual wavelength is set to 8 cm. Our NLOS imaging system captures data, reconstructs the dynamic hidden scene, and displays the result to the user live. The computational pipeline is acquisition-bound and easily supports a throughput of 5 fps with a latency of 1 second. The target SNR was set to be equal to the SNR at distance $z_0 = 1$ m. Hence, at the largest depth $z_{max} = 3$ m of our reconstruction we average three frames. The maximum motion velocity is given by the 200 ms exposure time and the 8 cm spatial resolution and is about 0.4 m per second, which is sufficient to capture normal human movements. Figure 4a shows the ground truth of the dynamic hidden scene and reconstructed NLOS frames are shown in Fig. 4b. Figure 4c shows the result after the depth-dependent frame averaging has been applied. More video frame results can be found in the Supplementary Results section. Additionally, readers are encouraged to view the

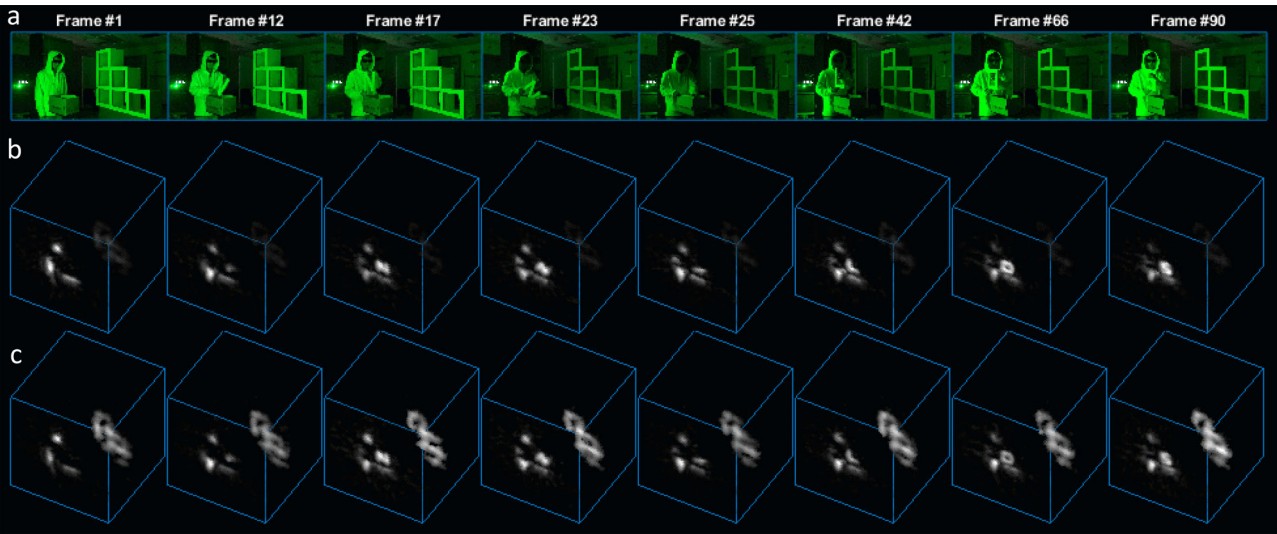

**Fig. 4 Live video: "NLOS letterbox".** Eight sample frames from a 20 s real-time video. The person takes four letters (N, L, O, S) out of a box. The first row shows ground truth images of the hidden scene. The second row shows reconstructed virtual frames using RSD. The third row shows reconstructed virtual frames after applying depth-dependent frame averaging.

full live NLOS videos in the Supplementary Video file, which shows the effectiveness of our reconstruction method in various scenarios including hidden mirrors and ambient light.

The authors affirm that persons appearing in Fig. 4 and Supplemental Videos have provided informed consent for publication of the images.

## Discussion

We have shown that the capabilities of NLOS imaging systems can be substantially increased with the combinations of purposefully designed SPAD arrays and array-specific fast reconstruction algorithms. Using a total of only 28 pixels we reconstruct live NLOS videos of non-retroreflective objects. We also show that scene size and object distance do not represent an insurmountable problem for NLOS imaging. While we believe that this is a major step forward in the demonstration of the capability of NLOS imaging and towards the actual deployment of NLOS imaging systems in real-world applications, such as robot navigation, disaster response, and many others, there are still opportunities for future work. CMOS SPAD array technology allows for the fabrication of kilopixel and recently even megapixel arrays at low cost[29]. We expect that future NLOS imaging systems will further improve capabilities by adding more pixels to improve SNR, speed, and stand-off distance, as well as increase the relay wall size to improve reconstruction resolution.

## Methods

**Principle of NLOS imaging and its consequences.** This section discusses the fundamental principle of NLOS imaging and its consequences on acquisition time and used laser power. In ToF NLOS imaging, a short laser pulse is sent towards a wall. Light from an illuminated spot on the relay surface scatters spherically in all directions. Some photons hit the objects in the hidden scene and are the origin of secondary spherical waves. After another reflection off of the relay wall, very few photons reach the detector and the roundtrip travel time of the photons, from the laser through the scene and back to the detector, is recorded. In this publication, we use Single-Photon Avalanche Diodes (SPADs) as detectors, arrays of which have become more and more available recently. A very illustrative animation of the NLOS imaging process can be found here[30]. The crucial step is how the light returns from the hidden scene objects to the relay wall: in the case of diffuse scene objects, the returning light fronts hit the full surface of the relay wall. This immediately makes it clear that it is beneficial to have a detector that covers the whole area to capture as many photons as possible. High-quality reconstructions require fine time resolution, e.g., 100 ps translate to about 3 cm in space, and the wall needs to be sampled with a spatial resolution on roughly the same scale, i.e., for the given time resolution a few centimeters. For this reason, it is optimal to have

one SPAD pixel look at an area of about 3 cm × 3 cm in the considered case, and all pixels combined cover the full visible area. This way, ignoring detector parameters such as sensor chip fill factor and focusing on geometry for now, the most photon efficient capture is realized. By contrast, if only one SPAD pixel looks at one point on the wall, a huge fraction of returning photons is lost. This has immediate consequences on the laser power. The power levels used so far are far from being eye-safe, which is a major concern. However, there is a direct relation between photon capture efficiency and used laser power, which will be described mathematically in the following. Please see Supplementary Note 3 for a detailed description of the underlying stochastics.

Due to Helmholtz reciprocity, we can scan a single-pixel SPAD throughout the wall and have a fixed laser position, or equivalently scan the laser and look at a stationary point with the SPAD, under the assumption that the laser beam area and the wall area covered by the single-pixel SPAD are of the same size. For this discussion, we proceed with a fixed laser and a scanning SPAD. The laser is pulsed, and for simplicity assume that the laser photon rate $\lambda$ is constant during each pulse. Let the expected value of photons emitted during pulse time $\Delta t$ be $\Lambda = \lambda \Delta t$. $L$ is the number of repeated laser pulses, and depending on the fixed laser repetition rate (not to be mistaken with the pulse duration), it obviously takes a certain fixed time to generate these $L$ pulses. Let us assume that the recorded temporal histograms have a bin width of $\Delta t$. For a stationary SPAD pixel always looking at the same point on the wall, depending on the geometrical distances and scene reflectance all lumped into the attenuation factor $\gamma(t)$, we expect

$$\Lambda_{\text{det}}(t) = \gamma(t)L\Lambda \tag{1}$$

photons in one specific time bin $t$. $\Lambda_{\text{det}}(t)$ can be zero if there are no objects at the scene voxels that had a round trip time $t$. A stationary SPAD pixel does not provide the required spatial information needed to provide 3D reconstructions, so we need to scan it across the wall. The rest of the argument is therefore straightforward: if we scan $Q$ points on the wall with one individual SPAD, the exposure time for one point will be reduced by the factor $Q$, meaning that the expected number of photons in one-time bin reduces to

$$\Lambda_{\text{det}}(t) = \frac{1}{Q}\gamma(t)L\Lambda . \tag{2}$$

However, if we use a SPAD array that simultaneously captures the histograms at all $Q$ relay wall positions, we are back at the original expected photon number (1), because each pixel now is illuminated for the total time needed to send the $L$ pulses. We are now capturing the light more efficiently by looking at all points simultaneously instead of just one point. Taking a closer look at (1), we see that reducing the laser photon rate in the array scenario by $Q$ makes us arrive at exactly the same number of photons as in the point scanning scenario (2):

$$\Lambda_{\text{det}}(t) = \gamma(t)L\frac{\Lambda}{Q} = \frac{1}{Q}\gamma(t)L\Lambda . \tag{3}$$

As a consequence, there's an inverse proportionality between the photon rate $\Lambda$ and therefore the laser power, and the number of SPAD pixels. Using $Q$ SPAD pixels allows for reducing the laser power by $Q$ while still maintaining the same Signal-to-Noise Ratio (SNR) in the measured histograms. If the SPAD array chip fill factor is not 100%, but say only 10%, the light efficiency is reduced by a factor of 10. But let's assume that in the future SPAD arrays with 10,000 pixels will be

available, so we still can reduce the laser power by a factor of 1000. We have seen that more pixels allow for proportionally reducing the laser power. Alternatively, one can use more pixels with the same laser power to increase the SNR of the captured data and subsequently the quality of the reconstructed video. The number of photons acquired in each histogram time bin follows a Poisson distribution (see Section 3.3 in the Supplementary Materials); for such a distribution, increasing the photon rate by $Q$ improves the SNR by a factor of $\sqrt{Q}$.

**Sparse illumination aperture remapping.** The notation in this section and the supplement is the same as in[17]; a list of symbols can be found in the supplement. We introduce a sparse relay wall illumination scanning pattern with a remapping operation based on SPAD array sensing points to create a virtual complete continuous illumination grid with a single sensing point. This approach has several benefits. First, the sparse illumination pattern reduces the physical scan time allowing to achieve a high capturing frame rate, which is essential when it comes to dynamic scenes. Second, the RSD method[17] assumes a continuous illumination grid $\mathbf{x}_p = (x_p, y_p, 0)$ and a single sensing point $\mathbf{x}_c = (x_c, y_c, 0)$. Therefore transient data collected from multiple sensing points $\mathbf{x}_{c,q} = (x_{c,q}, y_{c,q}, 0)$, $q \in [1, \dots, Q]$ using a SPAD array cannot be used directly. The remapping operation addresses this issue by virtually remapping the sparsely scanned grid $\bar{\mathbf{x}}_p$ and multiple sensing points $\mathbf{x}_{c,q} = (x_{c,q}, y_{c,q}, 0)$ into a complete illumination grid $\mathbf{x}_p = (x_p, y_p, 0)$ and a single sensing point $\mathbf{x}_{c,1} = (x_{c,1}, y_{c,1}, 0)$. The remapping operation exploits a spatial relationship between $\bar{\mathbf{x}}_p$ and the $\mathbf{x}_{c,q}$, and approximates the missing illumination positions which is illustrated in Fig. 2a. Mathematically, given a set of sparse laser positions $\bar{\mathbf{x}}_p$ and a set of SPAD positions $\mathbf{x}_{c,q}$, one seeks to obtain the measurement at a laser grid shifted by small amounts $\Delta\mathbf{x}_{c,q} = (\Delta x_{c,q}, 0, 0)$. This can be done by virtually remapping the captured data to shifted locations and approximating the required time responses as

$$H((\bar{\mathbf{x}}_p + \Delta\mathbf{x}_{c,q}) \to \mathbf{x}_{c,1}, t) \approx H(\bar{\mathbf{x}}_p \to (\mathbf{x}_{c,1} - \Delta\mathbf{x}_{c,q}), t), \quad q \in [1, \dots, Q]. \tag{4}$$

This holds as long as the length of the spatial shift $\Delta\mathbf{x}_{c,q}$ is small with respect to the distance between the laser position on the relay wall and the object location, and the object location and the SPAD position on the relay wall, respectively. The transient data acquired by the leftmost SPAD pixels are projected to the rightmost SPAD pixel location, which virtually creates a full illumination scan grid. Mathematically speaking, what we are interested in is the absolute difference between the actual, physical roundtrip path from the sparse laser position $\bar{\mathbf{x}}_p$ to the object at location $\mathbf{x}_v$ and back to one SPAD at $\mathbf{x}_{c,q}$, and the virtual path from the virtual laser location $\mathbf{x}_p$ to the object at $\mathbf{x}_v$ and back to the virtual SPAD position $\mathbf{x}_{c,1}$:

$$\begin{aligned} D := \Big| \Big( &\| \mathbf{x}_v - \bar{\mathbf{x}}_p \| + \| \mathbf{x}_v - \mathbf{x}_{c,q} \| \Big) \\ &- \Big( \| \mathbf{x}_v - (\bar{\mathbf{x}}_p + \Delta\mathbf{x}_{c,Q}) \| + \| \mathbf{x}_v - \mathbf{x}_{c,1} \| \Big) \Big| . \end{aligned} \tag{5}$$

Here, we have taken the maximum possible shift indicated by the capital $Q$ in $\Delta\mathbf{x}_{c,Q}$, and $\| \cdot \|$ denotes the Euclidean distance. As long as the difference $D$ is shorter than the spatial uncertainty introduced by the temporal system jitter, the approximation in (4) holds. For our actual system, as explained in the Hardware Section below, the temporal jitter is 85 ps, which translates to roughly 2.55 cm. Figure Fig. 2c graphically shows all possible values for $D$ for the scenario encountered in our experiments. The virtually remapped transient data $H((\bar{\mathbf{x}}_p + \Delta\mathbf{x}_{c,q}) \to \mathbf{x}_{c,1}, t)$ can fully utilize the fast RSD method allowing for live reconstruction:

$$\mathcal{I}(x_v, y_v, z_v) = \Phi(\mathcal{P}(\mathbf{x}_c, t)) = \Phi(\mathcal{P}(\mathbf{x}_p, t) \overset{*}{\underset{t}{}} H((\bar{\mathbf{x}}_p + \Delta\mathbf{x}_{c,q}) \to \mathbf{x}_{c,1}, t)), \tag{6}$$

where $\Phi(\cdot)$ is the wave propagation operator, $\mathcal{P}(\mathbf{x}_p, t)$ the Phasor Field illumination function starting at the relay wall and $\Phi(\mathcal{P}(\mathbf{x}_c, t))$ the Phasor Field function returning to the relay wall after propagating through the scene. The operator $\overset{*}{\underset{t}{}}$ denotes convolution with respect to time. Figure 2b shows 5 different reconstructions from 5 individual SPAD pixels using the full illumination pattern, and one reconstruction using the virtually remapped illumination pattern from the sparse illumination pattern with 5 SPAD array pixels. The reconstruction result from virtually remapped data has comparable quality to single-pixel reconstructions while allowing for significantly faster and more efficient data acquisition.

By consecutively scanning the relay wall with the sparse illumination grid, we get a sequence of reconstructions $\mathcal{V}(\tau) = \mathcal{I}(x_v, y_v, z_v; \tau)$, $\tau = [1, 2, \dots]$ with exposure time of $\Delta\tau$ seconds for each frame $\tau$. The frame rate of $\mathcal{V}(\tau)$ is $1/\Delta\tau$.

**Details on the depth-dependent frame averaging.** To compensate for the SNR decrease (Fig. 3a), we can apply linear depth dependent frame averaging. Note that one can choose the target SNR level arbitrarily. Without loss of generality, we choose this level to be the SNR at the distance $z_0$ that is closest to the relay wall. For the reconstruction slice $\mathcal{I}(x_v, y_v, z_v)$ at $z_v = z_0$ no averaging is needed. For all consecutive depths $z_i$, $i = [1, 2, 3, \dots]$, we take the average of the $N = \text{ceil}(z_i)$ past

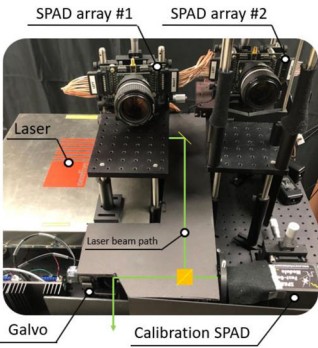

**Fig. 5 Hardware Setup.** Live NLOS Hardware setup layout and laser beam path scheme.

frames, that is,

$$\mathcal{I}_{avg}(x_v, y_v, z_i; \tau) = \frac{1}{N} \sum_{n=0}^{N-1} \mathcal{I}(x_v, y_v, z_i; \tau - n) . \tag{7}$$

For the Phasor Field method, we apply averaging to both the real and imaginary parts first before taking the absolute value squared (coherent summation). As a consequence, the statistical mean value of the averaged reconstruction does not equal the statistical mean of a single reconstruction, see Supplementary Figures 4 and 5.

In addition to the depth-dependent frame averaging we can compensate for the intensity decrease by applying depth-dependent intensity correction through multiplication by the mean values resulting from the SNR calculations (Supplementary Figures 4 and 5).

**Details on the hardware configuration, calibration, and acquisition.** The core components of our imaging system are the ultra-fast laser and the SPAD array. The used laser is a OneFive Katana HP pulsed laser operating at 532 nm with a pulse width of 35 ps. The operating laser power is 700 mW with a repetition rate of 5 MHz.

The prototype SPAD array has 16 pixels with a temporal resolution of about 50 ps FWHM and a dead time of 200 ns. The active gate window duration during which the SPAD is sensitive to photons can be adjusted. We set it to 40 ns, which corresponds to a round trip distance of roughly 6 m. In this work, we use two 16 by 1 pixel SPAD arrays[25], placed horizontally in a row in the imaging system, see Fig. 5. Both SPAD arrays are focused in the middle of the relay wall using Nikon 50 mm F1.2 objective lenses. Each pixel's observation area on the wall is approximately 5 mm². The width of the total SPAD array focus area on the relay wall is approximately 8 cm. One Thorlabs FL532-3 bandpass filter at 532 nm with FWHM 3 nm is placed in front of each SPAD array to reject ambient light of different wavelengths.

As a photon-counting device, we use the PicoQuant HydraHarp 400 Time-Correlated Single Photon Counting (TCSPC) unit with eight channels. Here we use the Time-Tagged Time-Resolved (TTTR) mode for the data acquisition at 8 ps time resolution. Combined, the effective temporal uncertainty of laser and each SPAD pixel is approximately 85 ps FWHM. One HydraHarp channel is used for the confocal single-pixel SPAD for system calibration. The remaining seven channels are used by both SPAD arrays; since the laser's repetition rate is 5 MHz, each HydraHarp channel has an available time window of 200 ns before the next laser pulse. Four pixels from the SPAD arrays are connected to one TCSPC channel. In order to separate the signals from the four SPAD pixels within a single TCSPC channel, we use cables of the corresponding length to delay the signals from each SPAD pixel. Thus, the 200 ns time window is divided into four sections: [0 - 40 ns], [40 - 90 ns], [90 - 140 ns] and [140 - 180 ns]. In total we utilize $7 \times 4 = 28$ SPAD pixels.

There's a tradeoff for the chosen virtual wavelength. The lower it is, the better the spatial resolution of the reconstructed scene. However, the wavelength is bounded from below: the temporal system resolution of about 85 ps translates to a full width at half maximum of roughly 2.5 cm in space. The virtual wavelength of the Phasor Field reconstruction cannot be smaller than twice this distance, otherwise, the reconstruction does not provide meaningful results. Given this time resolution, 8 cm provides a good tradeoff between low-noise, artifact-free reconstructability and high spatial reconstruction resolution. Furthermore, as the goal is to compose a virtual pulse in time that removes interference of targets at different depths[17], more frequency components and, correspondingly, virtual wavelengths need to be processed to compose this pulse, which is more time-consuming to calculate. Otherwise, if only one wavelength was selected for reconstruction, there would be a continuous wave travelling through the scene resulting in severe out-of-focus reconstruction artifacts. Please see[17] for more details on the implementation of PF reconstruction in the frequency domain.

We scan illumination points on the relay wall using a set of two mirror galvanometers (Thorlabs GVS012). The maximum frequency of this system is 150

| Acquisition | Method | Target distance | Exposure time | | | | | |
|---|---|---|---|---|---|---|---|---|
| | | | 0.2 sec | 1 sec | 4 sec | 60 sec | 10 min | 30 min |
| Non-confocal (Ours) | Ours | 1.2m | (img) | (img) | (img) | (img) | ✗ | ✗ |
| | | 2m | (img) | (img) | (img) | (img) | ✗ | ✗ |
| | Approx. LCT | 1.2m | (img) | (img) | (img) | (img) | ✗ | ✗ |
| | | 2m | (img) | (img) | (img) | (img) | ✗ | ✗ |
| | Approx. FK | 1.2m | (img) | (img) | (img) | (img) | ✗ | ✗ |
| | | 2m | (img) | (img) | (img) | (img) | ✗ | ✗ |
| Confocal | PF | 1.2m | ✗ | (img) | (img) | (img) | (img) | (img) |
| | | 2m | ✗ | (img) | (img) | (img) | (img) | (img) |
| | LCT | 1.2m | ✗ | (img) | (img) | (img) | (img) | (img) |
| | | 2m | ✗ | (img) | (img) | (img) | (img) | (img) |
| | FK | 1.2m | ✗ | (img) | (img) | (img) | (img) | (img) |
| | | 2m | ✗ | (img) | (img) | (img) | (img) | (img) |

**Fig. 6 Method Comparison.** Comparison with other methods.

Hz, meaning that 150 vertical lines per second can be scanned. The laser scans the relay wall continuously in a raster pattern. Our physical laser grid has 190x22 points. Vertical and horizontal spacing is 1 cm and 9 cm respectively. The full laser grid scanning rate is 5 fps and the exposure time per frame is 0.2 s, meaning that each laser point is exposed for 480 microseconds. Supplementary Figure 6 shows examples of data acquired by the SPAD array.

Our imaging system is located about 2 m away from the relay wall. To calibrate the system we use a single pixel SPAD that is aligned with the laser beam path (see Fig. 5). We scan the relay wall (1.9 m x 1.9 m) and the single pixel SPAD collects direct light from the relay wall, which yields the distances from the imaging system to the illumination points. Next, we scan a small square region around the SPAD array sensing points and collect the signal with SPAD array. The collected data is used to evaluate the distances from imaging system to the SPAD array sensing points on the relay wall. These distances are used to virtually shift the collected data to the relay wall.

The hidden scene starts at 1 m away from the relay wall and goes up to 3.5 m as limited by the time ranges provided by the TCSPC. The scene consists of conventional diffuse objects. The person in the scene is wearing a regular white hooded sweatshirt. Supplementary Figures 9, 10, 11, 12, 13 show examples of different scenes and reconstructions. These figures also contain depth dependent intensity corrected results. NLOS video results of corresponding scenes can be found in the supplementary video file.

**Comparison with other methods**. To demonstrate the effectiveness of our method, we provide a baseline comparison with other methods. Besides the two 16x1 non-confocal SPAD arrays, our imaging system has a single pixel gated SPAD that can be used for confocal data acquisition. The temporal resolution of the single pixel SPAD is approximately 30 ps. We capture the same scene using both the non-confocal and confocal SPADs. The non-confocal parameters are the same as described before; for the confocal measurement, we scan 128x128 laser illumination/detection points. Because of the limited vertical scan rate of the mirror galvanometers, the sparse non-confocal pattern can be scanned in 0.2 s, whereas almost 1 s is required to scan the full confocal grid. As a target we use a patch shaped as a "2" which has a white diffuse surface. The target was placed at distances of 1.2 m and 2 m in the hidden scene. For the non-confocal measurement, we captured the scene for an illumination time up to 60 s and for confocal measurement the scene was captured up to 1800 s.

Figure 6 shows the comparison which is split into two parts, non-confocal and confocal data.

The scene was captured using the non-confocal and confocal schemes. The non-confocal data are used for direct reconstruction with the proposed method, and are also approximately converted to confocal data with the method described in Lindell et al.[15], as also demonstrated in[17]. Then, the LCT[16] ("approximate LCT") and FK migration[15] ("approximate FK") are applied to this approximately confocal data. The reconstruction results of approximate LCT and approximate FK look blurry, have a hazy background, and look noisy at short exposure times. Our method successfully reconstructs the scene even with a short exposure time of 0.2 s. After 1 – 4 seconds of exposure the reconstruction quality doesn't change noticeably, which suggests our proposed method doesn't require long exposure times. Liu et al.[18] demonstrated that a Phasor Field reconstruction stabilizes at a certain exposure level and adding further exposure time does not affect the reconstruction. Using the confocal measurements we reconstruct the scene using the confocal Phasor Field reconstruction version[17], LCT[16] and FK[15]. The confocal acquisition method in general suffers from noise with short exposure times and requires a long exposure time to achieve high quality reconstruction of diffuse targets. Therefore, often times confocal methods use retroreflective materials to increase the signal. Since the confocal measurement requires a dense scan of the

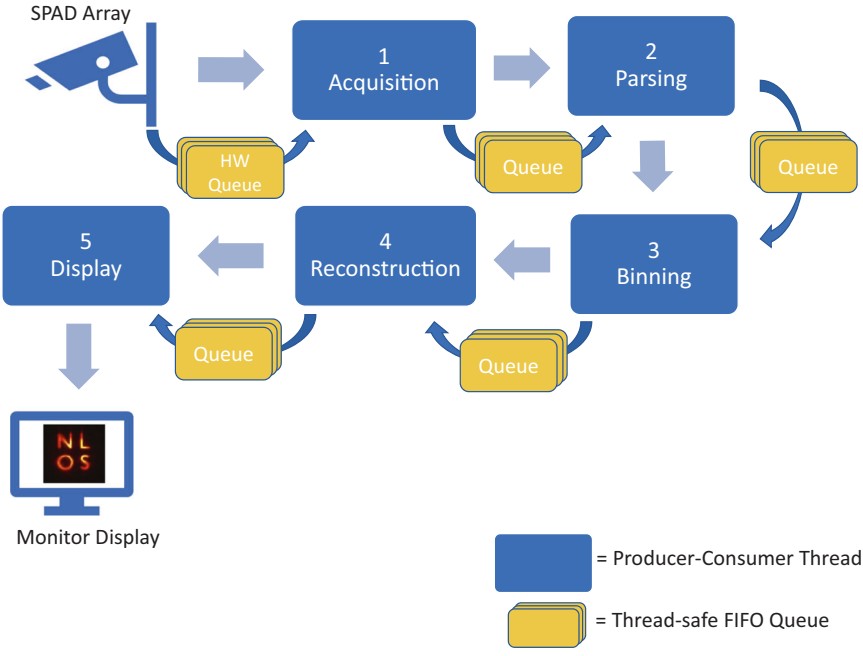

**Fig. 7 Software Block Diagram of Reconstruction Pipeline.** FIFO: First-In First-Out; SPAD: Single-Photon Avalanche Diode.

relay wall, a confocal scan at 0.2 seconds was not possible with our galvo mirrors. Note that the times reported here are in line with what was reported in the original papers.

**Details on the implementation of low-latency reconstruction pipeline**. Our custom software implements a processing pipeline that reads incoming photon data from the hardware, performs the NLOS image reconstruction, and displays resulting 2D images to the screen, keeping pace with hardware acquisition rates described above. The software is written in C++ and makes use of both CPU multithreading and GPU computation.

**Algorithm 1**. Frequency Domain Histogram Binning (un-optimized for clarity)
 **input :** Array, *photons*, of (*gridIdx*, *photonTime*) tuples
 Array, *freqs*, of frequencies
 **output:** Array *fdh*[NUM_FREQS][NUM_GRID_INDICES][2], Fourier Domain Histogram
 ZeroMemory (*fdh*);
 **for** $f = 0$ **to** *NUM_FREQS* **do in parallel**
 **foreach** (*idx*, *time*) in *photons* **do**
 *fdh*[*f*][*idx*][0] += SinLookupTable[(int)(*freqs*[*f*] * *time*)];
 *fdh*[*f*][*idx*][1] += CosLookupTable[(int)(*freqs*[*f*] * *time*)];
 **end**
 **end**
 **return** *fdh*;

The software is designed using a multi-stage producer-consumer model with the processing broken up into five distinct stages. Each stage runs in a separate thread on the CPU, and is connected to its predecessor and successor stages by thread-safe FIFO queues. Data travels through the stages sequentially, with raw photon event records entering the first stage, and 2D images exiting the final stage. Each stage's thread runs in an infinite loop, performing the same sequence of tasks: wait for data to become available, retrieve the available data from its incoming queue, process the data, and finally submit the processed data to its outgoing queue. Figure 7 shows a block diagram of the design. The five stages of processing are Acquisition, Parsing, Binning, Reconstruction, and Display. The staged pipeline and multithreaded model allows the entire pipeline to always remain full and working. That is, while frame $n$ is being displayed, simultaneously frame $n + 1$ is being reconstructed, frame $n + 2$ is being binned, frame $n + 3$ is being parsed, and photon events for what will become frame $n + 4$ are being collected. During properly tuned execution, the queues between stages never have more than a single entry waiting for processing, but serve primarily to decouple the processing of each stage. The implementation details of each stage are now described briefly.

**Acquisition** The first stage directly connects to the HydraHarp API to retrieve raw photon timing records in the T3 format (4 bytes per photon record) via the vendor-provided USB3 driver. In each iteration of its infinite loop, the thread polls the hardware driver for all available photon records that have accumulated in a driver-side queue. Beginning-of-frame and end-of-frame markers are encoded inline with the photon events in the T3 record format. After retrieving all available records from the hardware FIFO, the array of raw (unparsed) T3

records is passed to the next stage by enqueuing the records into this stage's outgoing queue, and the thread repeats its process of polling the hardware again.

**Parsing** The parsing thread retrieves arrays of raw photon records from its incoming queue and unpacks each T3 record into usable photon information. This includes de-multiplexing the SPAD channels, calculating the grid-index of the photon based on galvo time, adjusting photon arrival timing based on physical SPAD geometry, and searching for start-of-frame and end-of-frame markers. Having found these markers, this stage packages the newly-calculated tuples of (grid_index, photon_timing) into an array representing a single discrete image frame and submits this array to its outgoing queue.

**Binning** The binning thread receives an entire frame of pre-processed tuples containing the grid indices and timings of each photon's arrival. These records are binned directly into a frequency domain histogram (FDH). See Algorithm 1 for details. To increase performance, we use OpenMP to enlist all available CPU threads to perform the frequency for loop in parallel, as there are no data write hazards on this loop. We achieve good cache-coherency due to the chosen FDH memory layout. To further increase performance, we pre-multiply the frequency and omega values, and we discretize the photon arrival times to enable use of a pre-calculated lookup-table of sin and cos values that fits entirely within cache. The output of this stage is a FDH for a single frame.

**Reconstruction** The reconstruction thread dequeues a frame's FDH from its incoming queue and immediately transfers the FDH into GPU memory. Using an RSD kernel that has been pre-computed at application startup based on scene parameters, a sequence of CUDA kernels are executed to perform the Fast RSD algorithm's FFT, convolution with kernel, inverse FFT, and slice selection. The previous 3 reconstructed image cubes are held in memory and the resulting 2D image is formed by the depth dependent time averaging scheme described above. The resulting 2D image for this frame is then moved from GPU memory to main system memory and is enqueued in this stage's outgoing queue.

**Display** The final stages of the pipeline dequeues 2D images from its incoming queue. The 2D image is normalized, color-mapped, rotated, and scaled for display. The image is then displayed to the user, and the thread resumes waiting for the next 2D image to arrive in its incoming queue.

## Data availability

Example raw data files recorded using the described hardware setup are available at https://biostat.wisc.edu/~compoptics/rt_nlos21/rt_nlos.html.

## Code availability

Source code for the implementation used in this paper is available at https://biostat.wisc.edu/~compoptics/rt_nlos21/rt_nlos.html.

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

## Acknowledgements

This work was funded by DARPA through the DARPA REVEAL project (HR0011-16-C-0025), National Science Foundation grants NSF IIS-2008584, CCF-1812944, IIS-1763638, and IIS-2106768 and a grant from UW–Madison's Draper Technology Innovation Fund.

## Author contributions

A.V., J.H.N. and X.L. conceived the method, E.B., J.H.N. and E.S. implemented the low-latency pipeline. S.B. developed the SNR models. J.H.N. built the experimental setup and developed the simulations. J.H.N., M.R. and A.T. designed and built the SPAD array detectors and provided support for their integration in the system and operation., S.B. and A.V. analyzed the results. S.B. and X.L. helped with the system calibration. A.V. coordinated all aspects of the project. All authors contributed to the writing.

## Competing interests
SB and AV are shareholders of Ubicept. Ubicept develops imaging solutions with single photon cameras. SB is an employee of Ubicept and AV is a scientific adviser. The other authors declare no competing interests.
