## [Peer Review File · Nature Communications]

REVIEWER COMMENTS

Reviewer #1 (Remarks to the Author):

This paper proposes a method to achieve "real-time" NLOS imaging through SPAD. The paper extends prior work in the field with various contributions that enable faster performance.

Paper Strengths:

- + Experimental results show reconstruction of occluded scenes at real-time
- + The depth-dependent averaging of scene voxels allows some measure of depth invariance in the reconstructions (Equation 3).
- + There is a place for a real-time NLOS with SPAD paper in top journals like Nature Comms.

Paper Weaknesses (high-level):

- Positioning of the paper's premise. There are other real-time NLOS papers in the computational photography community. There are also other non-real-time NLOS papers that have higher quality. This paper appears to be at 5 FPS; it is not clear what tradeoffs were made and/or whether this is the key milestone.
- At this point, the NLOS field is becoming analogous to the deblurring field, in the sense that there is so much prior art one needs to show a collage of comparisons in the main paper. Since this paper stands on the shoulders of so much prior work: I would like the main paper to include baseline comparisons to show the tradeoffs wrt. prior art.
- Please characterize the tradeoffs in getting to real-time.
- Paper has some novelty, but is largely an extension of (16) with engineering/computational tricks to make it faster, without really characterizing the tradeoffs wrt the broader field.
- Paper title should be changed. The framerate should be specified, all experiments should be conducted at this framerate, and it should relate specifically to ToF in the title, since there are other NLOS methods.

Paper Weaknesses (detailed):

- In line 11, the manuscript writes "the first time captures and reconstructs real-time videos of non-line-of-sight scenes with natural non-retro-reflective objects". The statement is not necessarily incorrect, but could be slightly misleading as there are other techniques using, for example, thermal wavelengths, that obtain real-time imaging around corners.
- On that note, it is not clear why other forms of real-time NLOS are not referenced, such as Maeda et al. "Thermal NLOS" ICCP 2019. The use of more specular wavelengths for natural objects is another potential pathway to obtain real-time NLOS.
- line 21. "Robust reconstructions of near room-sized scenes have only been demonstrated with ToF". Please narrow the scope of this sentence. If a reconstruction is defined strictly to be in 3D, this statement is more correct. There are non-ToF methods that robustly recover 2D projections of the occluded scene.
- line 75. This sentence on noise should be double-checked, since there are quantum noise effects also added in traditional camera models, prior to image reconstruction.
- line 84: please add details on how the SPAD in (18) specifically is an improvement.
- line 100: what is the maximum object velocity? Consider adding to the main text.
- line 151: conventional real-time is 30 Hz, but exposure time of 5 fps is not precisely real-time. The paper should define why their approach at 5fps is considered real-time, and why we should publish this now rather than waiting for a truly real-time approach, at typical camera rates.
- line 164: it is hard to link "robustness of videos" to the videos. What sort of robustness are we looking for? Spatial frequency, temporal, artifact robustness etc?
- line 164: results look good, but it is not clear what the tradeoffs are wrt prior work that is not real-time. The results look worse than work out of the light cone transform line of papers and it is not clear how one should be benchmarking success.
- line 179: sparse remapping operation seems like an incremental extension on (16). Useful for increasing the throughput but not entirely unexpected to use a mapping function of multi-illumination grid and multi-sensing pixels.

Reviewer #2 (Remarks to the Author):

In this work, the authors significantly improve, in both hardware and software, the state-of-the-art approach to NLOS imaging, and achieve the impressive feat of real-time 3D imaging around the corner of realistic scenes, with reconstruction speed that is well beyond the current state-of-the-art. The main advancement is the use of an array of specially-designed SPAD detector, coupled with optimized spatiotemporal averaging and fast reconstruction algorithm. I am very impressed from the results that are achieved with realistic non-retro-reflecting objects and frame-rates, and I believe that the advancement is sufficiently important to justify publication in Nature Communications, although it may be considered mainly technical/engineering in nature. The reason for my recommendation to accept this work to NCOMMS (given the points mentioned below are addressed) is that this work may begin to persuade the previously skeptics (such as myself) on the applicability of optical NLOS imaging via ToF.

The manuscript and the supplementary are well written but there are several important aspects that must be addressed before this paper can be published:

- 1) The most important point is missing proof/justification for the aperture remapping process. The text claims this aspect is discussed in the supplementary but I could not find any explanation beyond the brief one in the Methods section. The approximation where the assumption is valid must be proven and explained. In addition, the concept of remapping, should be better explained both technically and its gain explained in the main text. A short explanation on the remapping should be given in the caption of Fig.2 as well.
- 2) An explanation for the lower than $1/r^4$ fall-off should be given in the main text, and not just in the supplementary
- 3) The remapped image in Fig.2 seems of lower quality than the 5 individual images, while it is claimed in the Methods that this image is of higher SNR. This aspect should be addressed in the text and Fig.2 caption
- 4) Line 87 claims that Phasor field and back-projection are not fast enough, but at the end the authors do use a phasor-field approach. I am guessing I have misunderstood the authors claim, so I deduce it can probably be phrased better.
- 5) How is RSD faster than phasor-field back-projection approach? Isn't it effectively the same? There is a need to expand the explanation on the reconstruction algorithms in the main text: what is backprojection what is RSD, what is the differences. I recommend adding the two in forms of equations in the main text. Otherwise, the reader has to look in the supplementary for the reconstruction algorithm used.
- 6) The authors use a rather high power laser for this work: what are the practical limits of performance for applying the proposed technique at eyesafe levels (with a possible wavelength change)
- 7) Eq.2 in the methods does not have the same form as the RSD equation in the supplementary. It would be easier to follow if they would have the same exact form.

Less critical points:

- 1) What is the optimal virtual wavelength source? how is it chosen? is there an advantage to utilize several wavelengths?
- 2) in ultrasound imaging, a similar problem of image reconstruction from TOF measurements is handled in real-time. Can fast beam forming algorithms from ultrasound imaging be employed for the NLOS problem?
- 3) "PF", "FDH", "RSD" appear in the caption of Fig.1 but are not explained. Same for BP in Fig.3
- 4) line 43 "Form"  "from"

I strongly support publication in NComms assuming the authors address my concerns above

Reviewer #3 (Remarks to the Author):

This work presents a well-crafted approach towards real-time Non-line-of-sight imaging. While most existing non-line-of-sight imaging approaches rely on single SPAD detectors combined with galvo scanning, the proposed method relies on an array of SPAD detectors. Specifically, the authors use a low-resolution detector array to spatially multiplex some scanning in the proposed capture approach.

The proposed method allows simultaneous multiplexed acquisition of non-line-of-sight data. The proposed reconstruction method is a modification of the Phasor Fields method that is not real-time per-se. The method may provide a path towards longer ranges, although the authors only show meter-size scenes.

The authors may also consider citing and discussing their work in context of:

Non-line-of-sight imaging over 1.43 km

Cheng Wu, Jianjiang Liu, Xin Huang, Zheng-Ping Li, Chao Yu, Jun-Tian Ye, Jun Zhang, Qiang Zhang, Xiankang Dou, Vivek K. Goyal, Feihu Xu, Jian-Wei Pan
Proceedings of the National Academy of Sciences Mar 2021, 118 (10) e2024468118; DOI: 10.1073/pnas.2024468118

Overall, the results demonstrate live non-line-of-sight reconstruction capabilities with good quality. As such, I am in support of publishing this work. However, major aspects of the work are incremental and the contribution is more on the systems level. Specifically, the very active field of Non-line-of-sight imaging has already shown

1) Real-time reconstruction and processing: At this point a number of works have tackled inference and capture speed. Chen et al. 2020 show a real-time reconstruction network, O'Toole recently demonstrated a fast acquisition approach purely based on galvo scanning, intensity-only methods by the Goyal group, Wornell's group and Torralba group are in some instances also real-time approaches (albeit at lower reconstruction quality). At the same time real-time acoustic and radar approaches have been demonstrated, recently as well by Schreiner et al. 2020 and Lindell et al. 2019.

2) SNR: While the proposed method acquires additional samples "for free" using the detector array, a key issue of most non-line-of-sight methods indeed remains unaddressed: the laser power. To truly make non-line-of-sight imaging more practical in the future, three orders of magnitude of laser power in industrial lasers have to be overcome. The proposed work uses the same high-power laser as existing works, from Liu et al. or Lindell et al.

3) Quality of experimental results: The quality of the experimental results is adequate but falls, obviously, short of the quality that recent non-real-time methods have been able to achieve. Given the very low image quality, the proposed method may be more convincing as a tracking approach, in which case the method also should compare against longer range tracking methods. Given that passive methods achieve 0-10 meters standoff distances, I remain unconvinced that the proposed method drastically improves on these existing methods for tracking applications.

4) Retroreflector claims: Existing approaches "engineer" their scenes to include highly reflective objects or shiny walls in the scene. A large number of works follow this somewhat unscientific approach. I would argue that the proposed method is not different in that sense. The person in the scene uses a highly reflective coat.

AUTHOR'S RESPONSE TO REVIEWERS

REVIEWER 1

Comment:

This paper proposes a method to achieve "real-time" NLOS imaging through SPAD. The paper extends prior work in the field with various contributions that enable faster performance.

Paper Strengths:

- Experimental results show reconstruction of occluded scenes at real-time
- The depth-dependent averaging of scene voxels allows some measure of depth invariance in the reconstructions (Equation 3)
- There is a place for a real-time NLOS with SPAD paper in top journals like Nature Comms.

Paper Weaknesses (high-level):

1.1) Positioning of the paper's premise. There are other real-time NLOS papers in the computational photography community. There are also other non-real-time NLOS papers that have higher quality. This paper appears to be at 5 FPS; it is not clear what tradeoffs were made and/or whether this is the key milestone.

Authors' response: Thank you for raising this very good point. The answer is covered by the new paragraph added to the introduction and discussed in more detail in the Methods section. Basically, one can add more SPAD pixels to cover a larger fraction of the relay wall with the detector (while still maintaining the spatial resolution; a single SPAD pixel looking at a large region does not provide sufficient information for 3D reconstruction), which allows to reduce the laser power proportional to the number of pixels.

Alternatively, one can keep the laser power constant when adding more pixels; in this case, the SNR of the data increases and subsequently the reconstruction quality. If each frame of the video had a longer acquisition time, say 2 s or 20 s instead of 0.2 s (for 5 fps), then the reconstruction quality improves and eventually reaches the one shown for example in Fig. 2 of Liu et al. , “Non-line-of-sight imaging using phasor-field virtual wave optics”, Nature, 572(7771):620–623, 2019. Please see Supplementary Figure 8 for an illustration of longer exposure times on the reconstruction quality, and also the newly added Figure 6 in the main paper. Furthermore, we added an explaining introductory paragraph to the Section “SNR considerations”.

1.2) At this point, the NLOS field is becoming analogous to the deblurring field, in the sense that there is so much prior art one needs to show a collage of comparisons in the main paper. Since this paper stands on the shoulders of so much prior work: I would like the main paper to include baseline comparisons to show the tradeoffs wrt. prior art.

Authors' response: We added the new Section “Comparison with other methods” to the Methods, where we compare the Phasor Field RSD reconstruction with the Light Cone Transform and FK migration, both on confocal and non-confocal data. Furthermore, we want to point out that fast confocal data acquisition so far only has been demonstrated with retroreflective objects, which are much brighter than the conventional diffuse objects we use. Please see also the newly added first section of the Supplement and Supplementary Figure 1, and our response to comment 3.4. Even if confocal data of diffuse objects are available in real-time, the SNR of the corresponding reconstructions is way worse than the non-confocal results we present here, since a confocal exposure time on the order of seconds is needed, instead of our non-confocal exposure time of 0.2 s (cf. Fig. 6).

1.3) Please characterize the tradeoffs in getting to real-time.

Authors' response: Please see our response to 1.1.

1.4) Paper has some novelty, but is largely an extension of (16) with engineering/computational tricks to make it faster, without really characterizing the tradeoffs wrt the broader field.

Authors' response: This comment mainly covers the computational aspect of our paper. However, we think that our essential contribution is the analysis of the SNR in NLOS imaging which has not been done before and subsequently the insight how the number of SPAD pixels affects the SNR and therefore reconstruction quality, which in turn allows us to create real-time videos. Furthermore, this insight also enables a reduction of the laser power to operate in the eye-safe regime. Both effects are major steps forward in the field and greatly add to the practical applicability of NLOS imaging. Actually presenting the real-time videos acquired with real SPAD array hardware underlines the correctness and importance of our findings.

1.5) Paper title should be changed. The framerate should be specified, all experiments should be conducted at this framerate, and it should relate specifically to ToF in the title, since there are other NLOS methods.

Authors' response: We changed the paper title accordingly.

Paper Weaknesses (detailed):

1.6) In line 11, the manuscript writes “the first time captures and reconstructs real-time videos of non-line-of-sight scenes with natural non-retro-reflective objects”. The statement is not necessarily incorrect, but could be slightly misleading as there are other techniques using, for example, thermal wavelengths, that obtain real-time imaging around corners.

Authors' response: We added “time-of-flight” to the respective line in the abstract to clarify. Please also see the shortcomings of other NLOS methods in the new real-time literature discussion section of the introduction.

1.7) On that note, it is not clear why other forms of real-time NLOS are not referenced, such as Maeda et al. “Thermal NLOS” ICCP 2019. The use of more specular wavelengths for natural objects is another potential pathway to obtain real-time NLOS.

Authors' response: We added a dedicated paragraph describing the real-time literature to the introduction and also included Maeda et al.’s paper.

1.8) line 21. “Robust reconstructions of near room-sized scenes have only been demonstrated with ToF”. Please narrow the scope of this sentence. If a reconstruction is defined strictly to be in 3D, this statement is more correct. There are non-ToF methods that robustly recover 2D projections of the occluded scene.

Authors' response: We added "3D" to the respective line.

1.9) line 75. *This sentence on noise should be double-checked, since there are quantum noise effects also added in traditional camera models, prior to image reconstruction.*

Authors' response: Thank you for the remark, we changed "reconstruction" to "formation" to clarify the explanation. With that change, it becomes clear that the camera noise is the last thing that affects the acquired image in a conventional camera, while in NLOS, a *noisy* measurement is acquired and then the image is calculated from the measurements.

1.10) line 84: *please add details on how the SPAD in (18) specifically is an improvement.*

Authors' response: We added the following text: "This is necessary because there are no commercial silicon SPAD arrays available yet which simultaneously fulfill all the requirements for good NLOS image reconstructions. Specifically, this array combines a high time resolution of less than 50 ps (FWHM) with the capability to independently read out each pixel individually to not miss any photon detection and the capability to gate the sensor, i.e., to have it inactive during the period in which the first bounce off the wall occurs. This is crucial for NLOS imaging to prevent this brightest return from overshadowing the subsequent dimmer signals from the hidden scene."

1.11) line 100: *what is the maximum object velocity? Consider adding to the main text.*

Authors' response: This is discussed in the last paragraph before the discussion: "The maximum motion velocity is given by the 200 ms exposure time and the 8 cm spatial resolution and is about 0.4 meters per second which is sufficient to capture normal human movements."

1.12) line 151: *conventional real-time is 30 Hz, but exposure time of 5 fps is not precisely real-time. The paper should define why their approach at 5fps is considered real-time, and why we should publish this now rather than waiting for a truly real-time approach, at typical camera rates.*

Authors' response: By real-time, we mean that the system is able to reconstruct the scene with minimum latency, not necessarily at 30 fps. In this context, we used the real-time definition of the Oxford English Dictionary <https://www.encyclopedia.com/caregiving/dictionaries-thesauruses-pictures-and-press-releases/real-time-imaging>: "real-time imaging (reel-tym) n. the rapid acquisition and manipulation of ultrasound information from a scanning probe by electronic circuits to enable images to be produced on TV screens almost instantaneously. Using similar techniques, the instant-

neous display of other imaging modalities, such as CT scanning and magnetic resonance imaging, can now be achieved”. Considering the updated manuscript text, we believe that our work is a major contribution to the field showing that such real-time imaging can actually be achieved; furthermore, we outline a path how such real-time imaging can be achieved at eye-safe laser light powers. This will greatly inform the further development of the field. Note the remarkable comment by Reviewer 2: ”The reason for my recommendation to accept this work to NCOMMS (given the points mentioned below are addressed) is that this work may begin to persuade the previously skeptics (such as myself) on the applicability of optical NLOS imaging via ToF”. Regarding the 30 fps frame rate raised as a concern, this requires about 6 times as many SPAD pixels. Given that the current array is a custom development which takes significant time effort to realize, we believe, in accordance with the above, that the researchers in the field should be informed about the great potential of ToF NLOS imaging as soon as possible in order to come up with advancements. Also note another comment by Reviewer 2: ”...with reconstruction speed that is well beyond the current state-of-the-art.”; we believe that such a major progress should be shared with the public.

1.13) line 164: it is hard to link “robustness of videos” to the videos. What sort of robustness are we looking for? Spatial frequency, temporal, artifact robustness etc?

Authors’ response: Thanks for pointing this out; by robustness, we meant the capability of our method to successfully reconstruct a wide variety of scenes under varying conditions such as ambient light. We replaced ”robustness” with ”effectiveness” to improve understandability.

1.14) line 164: results look good, but it is not clear what the tradeoffs are wrt prior work that is not real-time. The results look worse than work out of the light cone transform line of papers and it is not clear how

Authors’ response: Thank you for this comment. In response, we have added a comparison with other methods in the Methods section (”Comparison with other methods”). Please also see our response to comment 1.2. In short, even if the LCT methods provided better results (this is not the case, please see the newly added Figure 6), the issue with the confocal acquisition required by the LCT methods is that it only captures a fraction of the light that returns to the relay wall, so much longer exposure times are needed to nicely reconstruct diffuse objects. Existing confocal literature mostly images retroreflective objects, but these are not very common ”in the wild” and reflect much more light back to the sensing point on the wall than diffuse objects, see also the new Fig. 1 in the supplement.

1.15) line 179: sparse remapping operation seems like an incremental extension on (16). Useful for increasing the throughput but not entirely unexpected to use a mapping function of multi-illumination grid and multi-sensing pixels.

Authors' response: This is mainly the same concern as expressed in 1.4; please see our reply to that concern. The remapping operation can be seen as an intermediate step to deal with the low-resolution SPAD arrays currently available; once large-resolution arrays are available which cover the full relay wall, this step will not be needed anymore.

REVIEWER 2

Comment:

In this work, the authors significantly improve, in both hardware and software, the state-of-the-art approach to NLOS imaging, and achieve the impressive feat of real-time 3D imaging around the corner of realistic scenes, with reconstruction speed that is well beyond the current state-of-the-art. The main advancement is the use of an array of specially-designed SPAD detector, coupled with optimized spatiotemporal averaging and fast reconstruction algorithm. I am very impressed from the results that are achieved with realistic non-retro-reflecting objects and frame-rates, and I believe that the advancement is sufficiently important to justify publication in Nature Communications, although it may be considered mainly technical/engineering in nature. The reason for my recommendation to accept this work to NCOMMS (given the points mentioned below are addressed) is that this work may begin to persuade the previously skeptics (such as myself) on the applicability of optical NLOS imaging via ToF.

The manuscript and the supplementary are well written but there are several important aspects that must be addressed before this paper can be published:

2.1) The most important point is missing proof/justification for the aperture remapping process. The text claims this aspect is discussed in the supplementary but I could not find any explanation beyond the brief one in the Methods section. The approximation where the assumption is valid must be proven and explained. In addition, the concept of remapping, should be better explained both technically and its gain explained in the main text. A short explanation on the remapping should be given in the caption of Fig. 2 as well.

Authors' response: Sorry about the wrong reference to the supplement, it indeed should point to the Methods section. We fixed that and added a mathematical explanation to the part titled "Sparse illumination aperture remapping", please see the paper. In a nutshell, as long as the difference of the distances between physical scan points (laser and SPAD) and object and remapped scan points and object is smaller than the system jitter corresponding to about 2.55 cm, the remapping operation is valid. We also give a numerical example in Fig. 2c and improved this overall graphic, as well as added an explanation to the caption. We hope the whole method is better described now.

2.2) *An explanation for the lower than $1/r^4$ fall-off should be given in the main text, and not just in the supplementary*

Authors' response: We added this explanation to the SNR considerations section: "An intuitive explanation is this: imagine an infinitely large planar light source (in our case the relay wall illuminated by the laser) that emits a fixed light intensity into one direction. Everywhere in the corresponding half space we can measure the same intensity, which does not decrease with distance, otherwise, the total energy would change. A point object very close to a finite relay wall approximately fits this scenario, so the intensity loss from the relay wall to the object is almost negligible. Only the falloff from the object back to the wall has to be taken into account, which follows $1/r^2$."

2.3) *The remapped image in Fig. 2 seems of lower quality than the 5 individual images, while it is claimed in the Methods that this image is of higher SNR. This aspect should be addressed in the text and Fig. 2 caption*

Authors' response: Thank you for this comment. We want to clarify that using more SPAD pixels leads to an increased SNR in the reconstructions, but because of the spatial remapping operation, there can be minor artifacts in the reconstruction, when the approximation might be slightly violated. We have explained the remapping operation better both in the text ("Sparse illumination aperture remapping" in the Methods section) and visually in Fig. 2. In short: while more SPAD pixels help improve the SNR, the remapping operation necessary to create a full grid can introduce minor deterministic artifacts leading to slightly worse image quality.

2.4) *Line 87 claims that Phasor field and back-projection are not fast enough, but at the end the authors do use a phasor-field approach. I am guessing I have misunderstood the authors claim, so I deduce it can probably be phrased better.*

Authors' response: Thank you for pointing this out. We rewrote this paragraph to clarify, please see the revised manuscript. Essentially, Phasor Field reconstruction can be implemented in three different ways; the first two ones (direct integration and backprojection based) are too slow, and we use the third option, Rayleigh Sommerfeld Diffraction, which can be implemented efficiently in the Fourier domain. Its need for a continuously scanned laser grid is circumvented by the presented remapping approach.

2.5) *How is RSD faster than phasor-field back-projection approach? Isn't it effectively the same? There is a need to expand the explanation on the reconstruction algorithms in the main text: what is backprojection what is RSD, what is the differences. I recommend adding the two in forms of equations in the main text. Otherwise, the reader has to look in the supplementary for the reconstruction algorithm used.*

Authors' response: We hope that our response to the previous comment already clarifies most of the concern raised in this comment. To further emphasize that we only use the RSD Phasor Field reconstruction and non-PF backprojection as a benchmark, we added the sentence "Throughout this paper, we will only use Phasor Field reconstruction implemented as fast RSD for the real-time reconstructions, and Phasor field backprojection and conventional backprojection for the SNR comparisons." So there should not be any confusion. In addition, we refer the reader to Xiaochun Liu, Sebastian Bauer, and Andreas Velten, "Phasor field diffraction based reconstruction for fast non-line-of-sight imaging systems, Nature Communications, 11(1):1645, 2020", where the fast RSD and the difference with respect to backprojection/direct integration is explained in detail. Unless absolutely necessary, we would not give the respective equations in the main paper, as they only add more notational complexity and won't be used later. Also, we could give two versions; the time and space continuous one and the discrete one from the Supplement, but both wouldn't add to the paper understanding. We hope that the aforementioned discrimination between PF RSD and conventional backprojection in combination with the respective references prevents potential misunderstanding.

2.6) *The authors use a rather high power laser for this work: what are the practical limits of performance for applying the proposed technique at eyesafe levels (with a possible wavelength change)*

Authors' response: This is a very good point, see also Comment 3.2. We have described how the used laser power is inversely proportional to the number of used SPAD pixels in the newly added Methods section describing the general NLOS principle and its consequences. As a consequence, using more pixels (or equivalently exploiting a larger fraction of the area of the relay wall by looking at it with many detector pixels) allows to reduce the laser power, and the presented method utilizing a SPAD array is a first step in this direction. Larger SPAD arrays will allow to further reduce the laser power in the future and reach the eye-safe regime.

2.7) *Eq. 2 in the methods does not have the same form as the RSD equation in the supplementary. It would be easier to follow if they would have the same exact form.*

Authors' response: Thank you for pointing this out. We have fixed the notation in the main paper and added an additional line to Eq. 8 in the supplement, so the connection is made. In the supplement, we refer the reader to Xiaochun Liu, Sebastian Bauer, and Andreas Velten. Phasor field diffraction based reconstruction for fast non-line-of-sight imaging systems. Nature Communications, 11(1):1645, April 2020 for more details; repeating them in the supplement would require at least one more page of text which would distract from the high-level goal of deriving the SNR; these explanations already are very lengthy.

Less critical points:

2.9) *What is the optimal virtual wavelength source? how is it chosen? is there an advantage to utilize several wavelengths?*

Authors' response: We added an explanatory paragraph to the Methods subsection "Details on the hardware configuration, calibration and acquisition". We hope this paragraph is sufficient as a high-level explanation; more details can be found in the cited fast Phasor Field (RSD) paper.

2.10) *in ultrasound imaging, a similar problem of image reconstruction from TOF measurements is handled in real-time. Can fast beam forming algorithms from ultrasound imaging be employed for the NLOS problem?*

Authors' response: Yes, this is an interesting thought. We added a sentence to the introduction. Eventually, these methods will likely also rely on array detectors: In the context of NLOS imaging, the only paper using beamforming we are aware of is Pediredla, Adithya, Akshat Dave, and Ashok Veeraraghavan. "SNLOS: Non-line-of-sight scanning through temporal focusing." 2019 IEEE International Conference on Computational Photography (ICCP). IEEE, 2019, where this approach used to scan through each voxel of the hidden scene. However, this technique suffers from severe implementation issues. As the authors state: "In its raw form, SNLOS requires two-dimensional temporal delay profiles (delays required on the light rays hitting a location on the wall) to be implemented both on the illumination-side and on the imaging side. Moreover, these delays are required to be of the order of nanoseconds (for meter-scale scenes) making spatial light modulators incompatible to achieve these delays." Most importantly, this method uses two sets of galvos, one to scan the laser and one to scan a single pixel SPAD. This means that eventually, the beamforming method is also dependent on (SPAD) array detectors, since just looking at a single relay wall spot at a time with a single pixel SPAD simply wastes the vast majority of the light that returns to the wall (as the authors put it: "colossal photon loss", which by the way we are solving with the SPAD array approach). While the beamforming method does not require the application of a reconstruction method, our approach seems to be the method of choice, considering that it can be practically implemented in real time to reconstruct the full scene.

2.11) *"PF", "FDH", "RSD" appear in the caption of Fig. 1 but are not explained. Same for BP in Fig.3*

Authors' response: Thank you, good catch. These points have been fixed, please see the manuscript.

2.12) *line 43 "Form" □ "from"*
Authors' response: fixed

I strongly support publication in NComms assuming the authors address my

concerns above

REVIEWER 3

This work presents a well-crafted approach towards real-time Non-line-of-sight imaging. While most existing non-line-of-sight imaging approaches rely on single SPAD detectors combined with galvo scanning, the proposed method relies on an array of SPAD detectors. Specifically, the authors use a low-resolution detector array to spatially multiplex some scanning in the proposed capture approach.

The proposed method allows simultaneous multiplexed acquisition of non-line-of-sight data. The proposed reconstruction method is a modification of the Phasor Fields method that is not real-time per-se. The method may provide a path towards longer ranges, although the authors only show meter-size scenes.

The authors may also consider citing and discussing their work in context of:

Non-line-of-sight imaging over 1.43 km Cheng Wu, Jianjiang Liu, Xin Huang, Zheng-Ping Li, Chao Yu, Jun-Tian Ye, Jun Zhang, Qiang Zhang, Xiankang Dou, Vivek K. Goyal, Feihu Xu, Jian-Wei Pan Proceedings of the National Academy of Sciences Mar 2021, 118 (10) e2024468118; DOI: 10.1073/pnas.2024468118

Overall, the results demonstrate live non-line-of-sight reconstruction capabilities with good quality. As such, I am in support of publishing this work. However, major aspects of the work are incremental and the contribution is more on the systems level. Specifically, the very active field of Non-line-of-sight imaging has already shown...

Authors' response: Thanks for pointing out the long-range paper, we cited it in the introduction.

Comment:

3.1) Real-time reconstruction and processing: At this point a number of works have tackled inference and capture speed. Chen et al. 2020 show a real-time reconstruction network, O'Toole recently demonstrated a fast acquisition approach purely based on galvo scanning, intensity-only methods by the Goyal group, Wornell's group and Torralba group are in some instances also real-time approaches (albeit at lower reconstruction quality). At the same time real-time acoustic and radar approaches have been demonstrated, recently as well by Schreiner et al. 2020 and Lindell et al. 2019.

Authors' response: We added a dedicated paragraph describing the real-time literature to the introduction and also included papers from the mentioned groups. Please note that Lindell's acoustic paper (Lindell, Wetzstein, Koltun: Acoustic non-line-of-sight imaging, CVPR 2019) both from a signal acquisition and reconstruction perspective is not real-time: according to the paper, "The total chirp time at each scan position is therefore $16 \times 0.0625 \text{ s} = 1 \text{ s}$ and the total

scan time, including mechanical scanning, is approximately 4.5 min.” ... ”The iterative reconstruction requires 0.1 s per iteration without the LCT operator, and 9 s per iteration with the LCT operator and typically converges in several hundred iterations”. For this reason, we did not include this specific paper in the real-time literature overview, and refer to the more complete general overview of NLOS imaging (Faccio, Velten, Wetzstein, ”Non-line-of-sight imaging”, Nature Reviews Physics).

We are not sure which paper by O’Toole you have in mind. His papers such as Isogawa, Yuan, O’Toole, Kitani, ”Optical Non-Line-of-Sight Physics-based 3D Human Pose Estimation”, CVPR 2020 and the 2018 Nature paper on the Light Cone Transform (LCT) use retroreflective objects (or humans wearing retroreflective suits) for real-world experiments, which is covered by the sentence ”The only real-time ToF NLOS reconstructions to date, reconstruct retroreflective surfaces that for the specific scenes and using this specialized confocal scanning capture technique provide signals at least 10,000 times higher than diffuse surfaces in the demonstrated geometries and therefore are not indicative of NLOS performance in many real scenes” in our paper. The reason that confocal NLOS imaging tends to rely on retroreflective objects which return a lot of light is that this acquisition method wastes a lot of light, see also the newly added Methods section describing the general NLOS principle and its consequences, and our response to Comment 3.4.

3.2) SNR: While the proposed method acquires additional samples ”for free” using the detector array, a key issue of most non-line-of-sight methods indeed remains unaddressed: the laser power. To truly make non-line-of-sight imaging more practical in the future, three orders of magnitude of laser power in industrial lasers have to be overcome. The proposed work uses the same high-power laser as existing works, from Liu et al. or Lindell et al.

2.3) The remapped image in Fig. 2 seems of lower quality than the 5 individual images, while it is claimed in the Methods that this image is of higher SNR. This aspect should be addressed in the text and Fig. 2 caption

Authors’ response: This is a very good point. We have described how the used laser power is inversely proportional to the number of used SPAD pixels in the newly added introduction paragraph describing the general NLOS principle and its consequences. As a consequence, using more pixels to exploit a larger fraction of the area of the relay wall allows to reduce the laser power, and the presented method utilizing a SPAD array is a first step in this direction. Larger SPAD arrays will allow to further reduce the laser power in the future. Regarding the lower quality of the remapped image compared to the 5 individual images please see our response to Comment 2.3.

3.3) Quality of experimental results: The quality of the experimental results is adequate but falls, obviously, short of the quality that recent non-real-time methods have been able to achieve. Given the very low image quality, the proposed method may be more convincing as a tracking approach, in which case

the method also should compare against longer range tracking methods. Given that passive methods achieve 0-10 meters standoff distances, I remain unconvinced that the proposed method drastically improves on these existing methods for tracking applications.

Authors' response: The mentioned low quality is due to the low SNR of the data, which in turn is a consequence of the short exposure time needed to achieve 5 fps. The data SNR can be improved by adding more pixels and keeping the laser power constant. This is also described in the new introduction paragraph; please see the last sentences "Alternatively, one can use more pixels with the same laser power to increase the SNR of the captured data and subsequently the quality of the reconstructed video. The number of photons acquired in each histogram time bin follows a Poisson distribution (see Section 2.3 in the Supplementary Materials); for such a distribution, increasing the photon rate by Q improves the SNR by a factor of \sqrt{Q} ."

3.4) Retroreflector claims: Existing approaches "engineer" their scenes to include highly reflective objects or shiny walls in the scene. A large number of works follow this somewhat unscientific approach. I would argue that the proposed method is not different in that sense. The person in the scene uses a highly reflective coat.

Authors' response: Thank you for raising this very fundamental concern. Retroreflective materials reflect orders of magnitude more light than diffuse objects. This was already shown in Fig. 1 of the Supplementary Materials to O'Toole's Light Cone Transform paper (Matthew O'Toole, David B Lindell, and Gordon Wetzstein. Confocal non-line-of-sight imaging based on the light-cone transform. Nature, 555(7696):338, 2018). In our videos, the person was wearing two hooded sweatshirts with regular diffuse cloth surface: 1) GAP gray hoodie (link: https://www.gapfactory.com/browse/product.do?pid=510981021&cid=1112428&pcid=1112404&vid=1&grid=pds_14_29_1#pdp-page-content) and 2) amazon white hoodie (link: https://www.amazon.com/Amazon-Essentials-Fleece-Zip-up-Hoodie/dp/B07Q2GZSBK/ref=sr_1_9?crid=B9AET5SZDPKE&dchild=1&keywords=white+hoodie+amazon+essentials&qid=1622408346&srefix=white+hoodie+amazon). In order to compare the used objects with retroreflective ones, we set up an illustrative scene in a dark room shown in Fig. 1 of this document: it contains the two hooded sweatshirts and the stuffed toy used in the videos, as well as two retroreflective street signs and retroreflective tape. The top image is taken with room lights on and no camera flash, and the bottom image is taken with room lights off and camera flash. The latter resembles confocal NLOS acquisition, because the used cellphone camera has the flash and the camera aperture very close together, about 1.5 cm. As can be seen, the retroreflective objects send so much more light back to the camera (which mimics the relay wall) that the diffuse objects cannot be resolved by the camera and are completely dark, which shows that the objects we used indeed reflect very little light compared to retroreflective objects. We added this dis-

Figure 1: Scene containing the two hooded sweatshirts and the stuffed toy used in the paper. In addition, the scene has two retroreflective street signs and retroreflective tape. The images have been taken in a dark room. Top: image with room lights on, no camera flash; bottom: room lights off, camera flash.

cussion to the supplementary materials and refer to it in the main text.

REVIEWERS' COMMENTS

Reviewer #1 (Remarks to the Author):

As a professional courtesy, this review will be a bit direct. While the author response is detailed, and hits many of the points in previous reviews, I feel it unfortunately lags just a bit on two areas.

1. First, my concern about why 5 fps is a milestone still remains. As I mentioned in my review, 5 FPS is not the generally accepted definition of a real-time video device. The closest thing (in video imaging) we have to a community definition of "real time" is the framerate of the human eye. In the computational imaging community, this is often approximated at 30 Hz. One can simply google for this, but examples of real-time at 30 Hz include:

* <https://ieeexplore.ieee.org/document/854954>

* https://link.springer.com/chapter/10.1007/978-3-319-10590-1_5

Once it is outside the 30 Hz limit (which has precedence and justification), it becomes a bit curious why a 5 FPS video device is important enough for a venue like Nature Comms.

Where this reviewer had a bit of a disagreement was in the professional courtesy of the authors. The author response does not have a convincing technical argument or reference that explains why 5 FPS is an important goal. Instead, the author response tries to quote a dictionary definition from the Oxford English Dictionary, where, in this dictionary for non-specialists, "real-time" is taken to mean "almost instantaneously". Clearly, "almost instantaneously" does not help us advance the conversation. Quoting an English dictionary definition (for a non-specialist audience) to argue a specialist term, is among the more curious responses I've seen.

At this point, I think we have to be respectful to the limited cycles that referees, authors, and editors have in this pandemic. The authors should clearly explain why they decided to publish a paper at 5 FPS and why this number is of significance. Simply saying it's faster and a step toward future work (while making the quality compensations to get to 5 FPS) may or may not be a Nature Comms worthy result.

2. The authors have added a paragraph on the Maeda work; but the paragraph does not provide a more complete picture for a reader. For example, the Maeda work does not require "access to the hidden scene" for its typical use case of tracking objects that have a heat signature (e.g. a human, which is one of the major illustrations that this proposed work shows). However, what is lesser known is that lots of objects (not just humans, animals, but also pavement, concrete, foliage, etc) have different amounts of thermal emission that create contrast. Seen in this light, a wide variety of realistic objects/scenes would be able to be reconstructed in 2D form at 30 Hz. Of course, the problem is that: (a) it does not achieve optical contrast; (b) mostly 2D reconstruction; and (c) there are scenes where there wouldn't be any thermal emission. But in regards to (c), all tested scenes in the submitted paper would actually work for thermal emission). Seen in that light, the paper should have a much more balanced treatment of prior art, rather than suggesting only the downsides: requires "access to the hidden scene", and not the upsides (e.g. full 30 Hz performance, high resolution, etc). Although the Maeda paper is real-time (30 Hz), it is not in competition with this paper and the authors could perhaps try to provide a more balanced treatment to, as they perhaps hope to do, "lay a foundation for future work".

Once again, not just authors, but reviewers and editors have limited cycles and we must be fair to other papers as well. When a point is made in the reviews, please do try to empathize and consider the reviewer's viewpoint of a critique. Quoting back Oxford was seen by this reviewer, as a surface-level attempt to respond. Thanks for your kind understanding on this point.

Reviewer #3 (Remarks to the Author):

The authors responded to all of the concerns from the previous cycle. All of my concerns have been sufficiently addressed in the extensive response.

The only remaining concern is the comment regarding SNR that the authors bring up. As the authors mention, one needs to expose quadratically more (or use quadratically more pixels) to boost the SNR linearly. It would be wonderful if the authors could add a measurement at a large object-to-wall distance (e.g. 10m or 20m) in the final version. I would consider this optional, however. The paper is ready for publication in my opinion.

Reviewer's Comments and Author's Reply
for
“Real-time non-line-of-sight imaging at 5 frames per second”
(Final Revision)

Reviewer #1 (Remarks to the Author):

As a professional courtesy, this review will be a bit direct. While the author response is detailed, and hits many of the points in previous reviews, I feel it unfortunately lags just a bit on two areas.

1. First, my concern about why 5 fps is a milestone still remains. As I mentioned in my review, 5 FPS is not the generally accepted definition of a real-time video device. The closest thing (in video imaging) we have to a community definition of "real time" is the framerate of the human eye. In the computational imaging community, this is often approximated at 30 Hz. One can simply google for this, but examples of real-time at 30 Hz include:

* <https://ieeexplore.ieee.org/document/854954>

* https://link.springer.com/chapter/10.1007/978-3-319-10590-1_5

Once it is outside the 30 Hz limit (which has precedence and justification), it becomes a bit curious why a 5 FPS video device is important enough for a venue like Nature Comms.

Where this reviewer had a bit of a disagreement was in the professional courtesy of the authors. The author response does not have a convincing technical argument or reference that explains why 5 FPS is an important goal. Instead, the author response tries to quote a dictionary definition from the Oxford English Dictionary, where, in this dictionary for non-specialists, "real-time" is taken to mean "almost instantaneously". Clearly, "almost instantaneously" does not help us advance the conversation. Quoting an English dictionary definition (for a non-specialist audience) to argue a specialist term, is among the more curious responses I've seen.

At this point, I think we have to be respectful to the limited cycles that referees, authors, and editors have in this pandemic. The authors should clearly explain why they decided to publish a paper at 5 FPS and why this number is of significance. Simply saying its faster and a step toward future work (while making the quality compensations to get to 5 FPS) may or may not be a Nature Comms worthy result.

Author's Reply:

We apologize for the misunderstanding. We had hoped that stating the frame rate in the title of the paper, would remove uncertainty arising from vaguely defined terms. We are not aware of a formal definition of what framerate is required to create the appearance of a connected video, rather than a series of individual images, but for most individuals that transition appears to be

happening somewhere between 1 fps and 5 fps. In short, we feel justified in using the word video as long as we clearly and prominently state the frame rate.

The term “real time” in our title is not intended to refer to the frame rate, but to the fact that the video is available “live” without the need for extended processing. We apologize for this misunderstanding and are happy to change the term to “low latency” or whatever other term is preferred to clarify this.

Regarding the importance of the work: We agree that the particular frame rate of 5 fps has no particular significance. **Both 30 Hz and 5 Hz, or even 1 Hz are orders of magnitude higher than what had previously been reported.** We believe it is this dramatic increase in capture rate along with a straightforward path to increase it further, that makes this work significant.

In practice, the 5 Hz is limited by the speed of our galvo scanners and to a lesser extent the number of pixels in our array. Both are technical choices.

2. The authors have added a paragraph on the Maeda work; but the paragraph does not provide a more complete picture for a reader. For example, the Maeda work does not require "access to the hidden scene" for its typical use case of tracking objects that have a heat signature (e.g. a human, which is one of the major illustrations that this proposed work shows). However, what is lesser known is that lots of objects (not just humans, animals, but also pavement, concrete, foliage, etc) have different amounts of thermal emission that create contrast. Seen in this light, a wide variety of realistic objects/scenes would be able to be reconstructed in 2D form at 30 Hz. Of course, the problem is that: (a) it does not achieve optical contrast; (b) mostly 2D reconstruction; and (c) there are scenes where there wouldn't be any thermal emission. But in regards to (c), all tested scenes in the submitted paper would actually work for thermal emission). Seen in that light, the paper should have a much more balanced treatment of prior art, rather than suggesting only the downsides: requires "access to the hidden scene", and not the upsides (e.g. full 30 Hz performance, high resolution, etc). Although the Maeda paper is real-time (30 Hz), it is not in competition with this paper and the authors could perhaps try to provide a more balanced treatment to, as they perhaps hope to do, "lay a foundation for future work".

Once again, not just authors, but reviewers and editors have limited cycles and we must be fair to other papers as well. When a point is made in the reviews, please do try to empathize and consider the reviewer's viewpoint of a critique. Quoting back Oxford was seen by this reviewer, as a surface-level attempt to respond. Thanks for your kind understanding on this point.

Author's Reply:

Upon re-examining the paragraph, we agree that we could do a better job at summarizing the key points of the paper. We apologize for this oversight. We have rewritten the paragraph and believe it now better explains the key points of Maeda et. al. and its relation to our work.

Reviewer #3 (Remarks to the Author):

The authors responded to all of the concerns from the previous cycle. All of my concerns have been sufficiently addressed in the extensive response.

The only remaining concern is the comment regarding SNR that the authors bring up. As the authors mention, one needs to expose quadratically more (or use quadratically more pixels) to boost the SNR linearly. It would be wonderful if the authors could add a measurement at a large object-to-wall distance (e.g. 10m or 20m) in the final version. I would consider this optional, however. The paper is ready for publication in my opinion.

Author's Reply:

This is an excellent suggestion. Unfortunately, we are limited by our current hardware constraints. Our multiplexed TDC channels can only bin about 50 ns worth of data limiting our depth range to about 3 meters. This would in principle still allow us to reconstruct a volume from 10 m to 13 m out. However we currently don't have access to a large enough room that is classified as a laser lab to permit the operation of non eye-safe equipment. The logistical challenges of obtaining permission to operate the laser in a hallway or lecture room make it hard to complete this measurement in reasonable time. We are working hard to make long range experiments possible in future work either by gaining access to an appropriate space or by making the system eyesafe through faster scanning.